# Visual Whole-Body Control for Legged Loco-Manipulation

**Minghuan Liu**\*, **Zixuan Chen**\*, **Xuxin Cheng**, **Yandong Ji**
**Rizhao Qiu**, **Ruihan Yang**, **Xiaolong Wang**
UC San Diego
https://wholebody-b1.github.io/

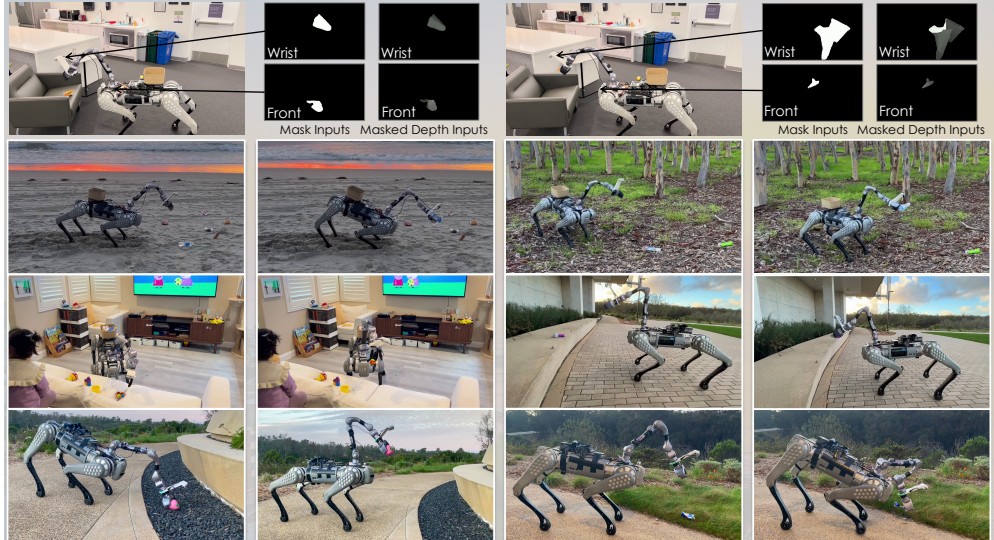

Figure 1: **Real-Robot input visualization and grasping trajectory.** Our framework enables the robot to grasp different objects in varying heights and surroundings. Our policy is trained in simulation using visual inputs (masks and masked depth images from two views) and can be deployed in real robots without any real-world data collection or fine-tuning.

**Abstract:** We study mobile manipulation using legged robots equipped with an arm, namely legged loco-manipulation. The robot legs, while usually utilized for mobility, offer an opportunity to amplify the manipulation capabilities by coordinating arms for whole-body control. We propose a framework that can conduct the whole-body control autonomously with visual observations. Our approach, namely Visual Whole-Body Control (VBC), is composed of a low-level control policy using all degrees of freedom to track the body velocities along with the end-effector position, and a high-level task-planning policy proposing the velocities and end-effector position based on visual inputs. We train all policies in simulation and perform Sim2Real transfer for real robot deployment. Extensive experiments show clear advantages over baselines in picking up diverse objects in various configurations (heights, locations, orientations) and environments.

**Keywords:** Robot Learning, Whole-Body Control, Legged Loco-Manipulation

## 1 Introduction

The study of mobile manipulation has achieved large advancements with the progress of better manipulation controllers and navigation systems. While installing wheels for a manipulator can help solve most household tasks [1, 2], it is very challenging to adopt these robots outdoors with challenging terrains. Imagine going camping, having a robot picking up trash and wood for us

8th Conference on Robot Learning (CoRL 2024), Munich, Germany. *Equal Contribution.

would be very helpful. To achieve such flexibility and applications in the wild, we study mobile manipulation with legged robots equipped with an arm, i.e., legged loco-manipulation. Specifically, we are interested in how the robot can conduct tasks using visual observation *autonomously*.

An important aspect of loco-manipulation is the opportunity to use all degrees of freedom (e.g., 19 DoF in this paper) of the robot for whole-body control. While the legs are typically utilized for mobility, there is a large potential for coordinating legs and arms together to amplify manipulation capabilities and workspace. We humans do this, too: we stand up to reach books placed on a shelf; we bend to tie our shoes. Fig. 1 shows tasks our robot can do by such a joint control. For example, our robot can go to the side of the road and pick up the trash on the grass. This will be unachievable without coordinating the legs of the robot, as the arm alone can hardly reach the ground.

While appealing, it is a very challenging control problem. The robot will need to exploit the contact with its surroundings and the objects and maintain stability and robustness to external disturbances all at the same time. Besides, acquiring a mobile platform to do precise manipulation jobs requires a steady robot body, which is much harder for legged robots compared to wheeled ones. An even larger challenge comes from achieving all these in diverse environments autonomously, given only the observation from an egocentric camera. Recent visual-learning-based approaches have shown promising results of locomotion on legged robots, like avoiding obstacles, climbing stairs, and jumping over stages [3, 4, 5, 6]. However, loco-manipulation requires more precise control, and the extra complexity can make direct end-to-end learning infeasible.

In this paper, we introduce a two-level framework for loco-manipulation, where a high-level policy proposes end-effector pose and robot body velocity commands based on visual observations and a low-level policy tracks these commands. Such a low-level controller is general for different tasks and the high-level policy is task-relevant. We named our framework **V**isual Whole-**B**ody **C**ontrol (VBC). Specifically, VBC contains three stages of training: first, we train a universal low-level policy using reinforcement learning (RL) to track any given goals and achieve whole-body behaviors; then, we train a privileged teacher policy by RL to provide immediate goals and guide the low-level policy to accomplish a task (e.g., pick-up); finally, we distill the teacher policy into a depth-image-based visuomotor student policy via online imitation learning. All training is done in simulation, and we perform Sim2Real transfer to deploy policies on real robots with zero human-collected data.

Our hardware platform is built on a Unitree B1 quadruped robot equipped with a Unitree Z1 robotic arm. We validate the effectiveness of each designed module of VBC in simulation on various objects and heights. As for the real-world experiments, we conducted the pickup task with 14 different objects, including regular shapes, daily uses, and irregular objects, under three height configurations, i.e., on the ground, on a box, and on a table, achieving a high success rate with emergent retrying behaviors and generalization ability in all scenarios.

## 2 Related Work

A set of works has exploited legged loco-manipulation to take advantage of legged robots. Bellicoso et al. [7] designs an online motion planning framework, together with a whole-body controller based on a hierarchical optimization algorithm to achieve loco-manipulation. Fu et al. [8] proposed an effective method to train a whole-body controller end-to-end through RL. Ma et al. [9] combined model-based manipulator control and learning-based locomotion, but only showed the response to external push instead of achieving manipulation tasks. Notably, these works all require human involvement and teleoperation to execute and achieve tasks, rather than operating autonomously. Zhang et al. [10], Yokoyama et al. [11], Arcari et al. [12] all performed an autonomous mobile manipulation system on quadruped robots while they utilized the default model-based controller provided through the robot APIs (only control robot locomotion in a static height) and only focused on the manipulation part without any whole-body behaviors to adapt to different heights of objects. Ferrolho et al. [13] proposed a trajectory optimization framework based on a robustness metric for solving complex loco-manipulation tasks, but the task is assigned and the target positions are calculated via a motion capture system, thus only works in limit. Sleiman et al. [14] realized a

bilevel search strategy for motion planning, by incorporating domain-specific rules, combing trajectory optimization, informed graph search, and sampling-based planning, enabling robots to perform complex tasks. They require the user to provide a dense description of the scene to initialize the solver. Zimmermann et al. [15] also utilized trajectory optimization to enable a legged robot to walk and pick up objects. However, they only show limited surroundings and simple objects. Besides these works with arm-based configuration, Wolfslag et al. [16] use the legs for pushing and box lifting tasks by introducing additional support legs named prongs to enhance the robustness and manipulation capabilities of quadrupedal robots. Jeon et al. [17] adopted a similar hierarchical framework to us, but does not involve any visual inputs, and only learned a quadruped for pushing large objects without an additional arm.

The system we design highly differs from the aforementioned works, as our work is a *autonomous* and vision-based loco-manipulation system on quadruped robots, with very limited human annotation before start. The robot is controlled and planned by end-to-end policies integrated with perception and control, able to adapt to objects with varying heights through its whole-body behavior, and emerges retrying behaviors when fails. Due to our design of visual inputs, our system can work both indoors and outdoors without external limits, as illustrated in Fig. 1, compared with almost all of the above-mentioned works that are only tested in limited scenarios. Fig. 2 shows why VBC is highly advantaged of static-height methods [10, 11, 12], for the body behavior allowing much more flexible poses (e.g., bending) to reach objects in random heights, like the one on the ground.

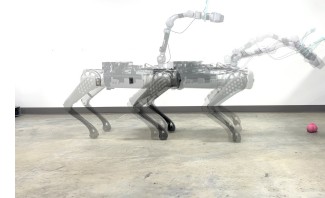

(a) Static-height.

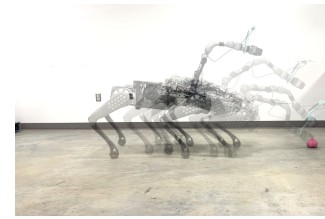

(b) VBC.

Figure 2: Illustration of comparing VBC to static-height methods. Static-height methods cannot reach objects on the ground without arm-leg coordination.

## 3 Visual Whole-Body Control

The proposed VBC framework (Fig. 3) consists of a low-level goal-reaching policy and a high-level task-planning policy. The low-level policy tracks a root velocity command and a target end-effector pose. The high-level policy provides these commands to the low-level policy, based on segmented depth images and proprioception states. When combined, our robot can achieve loco-manipulation tasks autonomously. This section draws details of the proposed framework.

### 3.1 Low-Level Policy for Whole-Body Goal-Reaching

The low-level control policy $\pi_{\text{low}}$ that takes over the whole-body control as described in the blue part in Fig. 3, is trained to track any given end-effector pose and body velocity across various terrains. We train such a policy with RL to realize a universal and robust behavior that can become the foundation of any high-level tasks.

**Commands.** The command $\mathbf{b}_t$ for our low-level policy is defined as $\mathbf{b}_t = [\mathbf{p}^{\text{cmd}}, \mathbf{o}^{\text{cmd}}, v_{\text{lin}}^{\text{cmd}}, \omega_{\text{yaw}}^{\text{cmd}}]$ where $\mathbf{p}^{\text{cmd}} \in \mathbb{R}^3, \mathbf{o}^{\text{cmd}} \in \mathbb{R}^3$ are end-effector position and orientation[1] command, defined under a height-invariant robot frame (details in Appendix C.1); $v_{\text{lin}}^{\text{cmd}} \in \mathbb{R}, \omega_{\text{yaw}}^{\text{cmd}} \in \mathbb{R}$ are the desired forward linear (x-axis) and yaw velocity under robot base frame, respectively. During the low-level policy training, we uniformly sample the linear velocity from the range $[-0.6\text{m/s}, 0.6\text{m/s}]$ and robot yaw velocity from $[-1.0\text{rad/s}, 1.0\text{rad/s}]$. When solving downstream tasks, the commands are specified by the high-level task-planning policy and can be further clipped into smaller ranges as needed.

**Observations.** The observation of our policy is formalized as a 90-dimensional vector $\mathbf{o}_t = [\mathbf{s}_t^{\text{base}}, \mathbf{s}_t^{\text{arm}}, \mathbf{s}_t^{\text{leg}}, \mathbf{a}_{t-1}, \mathbf{z}_t, \mathbf{t}_t, \mathbf{b}_t]$. Among them, $\mathbf{s}_t^{\text{base}} \in \mathbb{R}^5$ is the current quadruped base state including row, pitch, and yaw velocities, $\mathbf{s}_t^{\text{arm}} \in \mathbb{R}^{12}$ is arm state (position and velocity of each arm joint except the end-effector), $\mathbf{s}_t^{\text{leg}} \in \mathbb{R}^{28}$ is leg state (joint position and velocity of each leg joint,

---

[1]We use Euler angle parameterization by default in this paper.

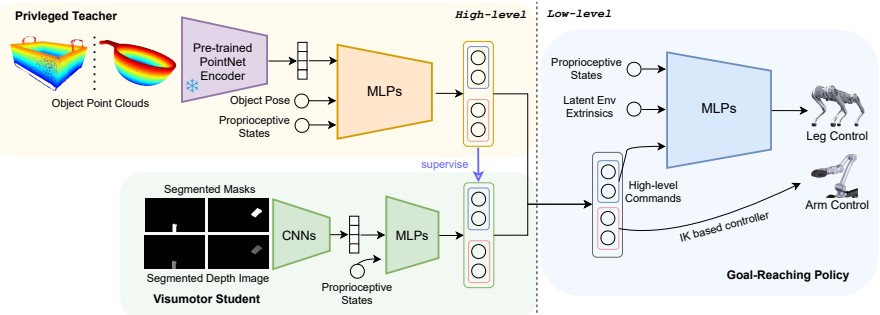

Figure 3: **The proposed VBC framework for loco-manipulation**. VBC trains a low-level control policy and a high-level planning policy using reinforcement learning and imitation learning.

and foot contact patterns), $\mathbf{a}_{t-1} \in \mathbb{R}^{12}$ is the last output of policy network, $\mathbf{z}_t \in \mathbb{R}^{20}$ is environment extrinsic vector, which encodes environment physics [8]. Following the work of Agarwal et al. [4], we aim to learn a steady walking behavior, such as trotting. To achieve this, we incorporate additional timing reference variables, denoted as $\mathbf{t}_t$, into the observation. These variables are computed from the offset timings of each foot by repeating the gait cycle of the desired symmetric quadrupedal contact pattern.

**Actions.** Our low-level goal-reaching policy mainly controls the quadruped robot. For the quadruped robot, our low-level policy outputs the target joint angles for all 12 leg joints, which is converted to torque command by a PD controller.

**Arm control.** Previous work [8] has revealed that whole-body RL policy struggles in controlling accurate end-effector position and orientation by controlling arm joint positions, which is critical for our manipulation tasks. Hence, we predict the end-effector pose command and convert it into target joint angles by solving the inverse kinematic (IK) with the pseudoinverse Jacobian method[2]. Formally, IK computes the increment of the current joint position $\Delta\theta \in \mathbb{R}^6 : \Delta\theta = J^T \left( JJ^T \right)^{-1} \mathbf{e}$, where $J$ is the Jacobian matrix, and $\mathbf{e} \in \mathbb{R}^6$ is the difference between the current end-effector pose and the next end-effector pose represented as the position and the Euler angle.

**End-effector goal poses sampling.** To train a robust low-level controller, we sample and track various gripper poses. To make sure we can sample smooth trajectories of gripper target poses for tracking, we employ a height-invariant coordinate system originating at the robot's base and randomly sample goals in this frame at every fixed timestep. This helps to learn a whole-body behavior and enables an expanded workspace. Details can be referred to Fig. 9 in Appendix C.1.

Due to the page limit, we leave more details of low-level policy training (rewards, policy architecture, goal sampling, randomization, etc.) in Appendix C.1.

## 3.2 High-Level Policy for Task Planning

When deploying our policy into the real world, the robot only perceives the object through depth images to achieve high-frequency control. However, RL with visual observation is quite challenging and requires particular techniques to be efficient [18, 19, 20, 21, 22]. To achieve simple and stable training, we train a privileged state-based policy that can access the shape and pose information of the object (the orange part of Fig. 3) and distill a visuomotor student (the green part of Fig. 3).

### 3.2.1 Privileged Teacher Policy

Like the low-level policy, the privileged teacher policy is trained through RL using task-specific reward functions and domain randomization, with a fixed pre-trained low-level policy underlying.
**Privileged observations.** The privileged observations include the latent shape features and object poses. We formally define it as a 1094-dim vector: $\mathbf{o}_t = [\mathbf{z}^{\text{shape}}, \mathbf{s}_t^{\text{obj}}, \mathbf{s}_t^{\text{proprio}}, \mathbf{v}_t^{\text{base vel}}, \mathbf{a}_{t-1}]$, where

---

[2]Other advanced IK solvers can be exploited for better performance under certain circumstances, but the pseudoinverse Jacobian method works well enough for the problem studied in this paper.

$\mathbf{z}^{\text{shape}} \in \mathbb{R}^{1024}$ is the latent shape feature vector encoded from the object point clouds using a pre-trained PointNet++ [23], which is fixed and $\mathbf{z}^{\text{shape}}$ remains invariant during training. $\mathbf{s}_t^{\text{obj}} \in \mathbb{R}^6$ is the object pose in local observation w.r.t the robot arm base. It contains the local pose of the object. The use of local coordination inherently incorporates the information regarding the distance to the object. $\mathbf{s}_t^{\text{proprio}} \in \mathbb{R}^{53}$ consists of joint positions including gripper joint position $\mathbf{q}_t \in \mathbb{R}^{19}$, joint velocity not including gripper joint velocity $\dot{\mathbf{q}}_t \in \mathbb{R}^{18}$, and end-effector position and orientation $\mathbf{s}_t^{\text{ee}} \in \mathbb{R}^6$. $\mathbf{v}_t^{\text{base vel}} \in \mathbb{R}^3$ is the base velocity of robot and $\mathbf{a}_{t-1} \in \mathbb{R}^9$ is the last high-level action.

**High-level Actions.** The high-level policy $\pi^{\text{high}}$ outputs actions that determine the velocities and the target end-effector position command for the low-level policy: $\mathbf{a}_t = [\mathbf{p}_t^{\text{cmd}}, \mathbf{v}_t^{\text{cmd}}, p_t^{\text{gripper}}] \in \mathbb{R}^9$, where $\mathbf{p}_t^{\text{cmd}} \in \mathbb{R}^6$ is the increment of gripper pose, $\mathbf{v}_t^{\text{cmd}} \in \mathbb{R}^2$ is the command of quadruped linear velocity and yaw velocity and $p_t^{\text{gripper}} \in \{0, 1\}$ indicates the gripper status (close or open).

**Domain randomization.** When training our high-level policy, we randomized the friction between the robot and the terrain, the mass, and the center of mass of the robot. Although the low-level policy already handles some of these gaps, these randomizations help the high-level policy adjust well to different low-level behaviors. Moreover, due to the uncertainty of the high-low signal transmission in the real world, we also randomize the high-level frequency by involving random low-level policy calls during one high-level call. We also randomize the Kp and Kd parameters for the arm controller. Regarding the pick-up task, we randomize table heights, the initial position and pose of the robot, along with the position and pose of objects.

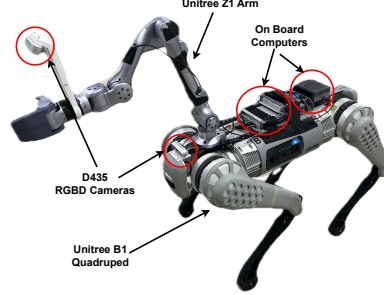

Figure 4: **Real robot system setup.**

### 3.2.2 Visuomotor Student Policy

We find robust and adaptive picking-up behaviors emerge from the privileged teacher policy training, with inaccessible object poses and shape features in reality. To deploy our policy in the real world, we must use easily accessible real-world observations like images.

**Observations and actions.** Although camera images are easily accessible both in simulation and the real world, there is a huge gap for colored images between them. As the main difference and noise come from the surrounding environment in simulated and real images, we turn to segmented depth images to train the visuomotor policy. Formally, the observation of our vision policy is $\mathbf{o}_t = [\mathbf{o}_t^{\text{image}}, \mathbf{s}_t^{\text{proprio}}, \mathbf{a}_{t-1}]$, where $\mathbf{o}_t^{\text{image}}$ consists of object segmentation masks and segmented depth images. $\mathbf{s}_{\text{proprio}}$ and $\mathbf{a}_{t-1}]$ are proprioceptive observation and last action defined above. The output actions are the same as those of the teacher policy.

**Imitation learning for student distillation.** To deploy our policy in the real world, we distill our state-based high-level planning policy into a vision-based policy using DAgger, which warms up the training by teacher sampling the initial transition data. Then we sample transitions using the student policy and request correction from the teacher. We find this is much more efficient and effective than directly learning a visuomotor student policy using RL.

**Randomization and augmentation.** To bridge the sim-and-real gap, we slightly randomize the position and rotation of both cameras to handle slight offsets in real. Besides, we clip the depth image to a minimum of 0.2m to avoid low-quality images when the distance is too close to a real camera, after which we normalize the depth value. We adopt augmentations like RandomErasing, Gaussian-Blur, GaussianNoise, and RandomRotation for the depth images. Additional details regarding the high-level policy training including rewards can be found in Appendix C.2.

### 3.3 Real World Deployment

**Robot Platform** The robot platform used in this work comprises a Unitree B1 quadruped robot with a Unitree Z1 robot arm mounted on its back. Our robot equips two cameras, one on the head and another close to the gripper. We use the onboard computer of B1 to control the low-level policy, and an additional onboard computer to receive visual inputs and control the high-level policy. The Unitree B1 quadruped has 12 actuatable DoFs, the arm has 6 DoFs, and the gripper has 1 DoF, making 19 DoFs in total. An overview of the hardware setup is shown in Fig. 4.

**Vision masks.** Note that our system works with little human intervention, i.e., we need to annotate and segment objects from environment backgrounds. To this end, we take use of TrackingSAM [24] (details in Appendix J), a tool that combines a video object tracking model [25] with interactive segmentation interface [26]. The implementation details and an example for illustration are given in Sec. J and Fig. 14. During real-world experiments, we annotated the objects from one of the RGBD cameras at the beginning of each reset trial. TrackingSAM will keep tracking the segmented object and providing the mask in real time (∼10 fps on the robot). The depth images obtained from the RGBD cameras are pre-processed by hole-filling and clipping. Empirically, as long as the object is visible to one camera (either the head or the wrist camera), the annotation mask can be propagated to both views to initialize an automated pickup.

## 4 Experiments

We conduct a set of experiments on pick-up tasks to compare VBC with baselines both in simulation and the real world, showing the effectiveness of training a loco-manipulation policy with VBC, and the generalizability over varying objects with different heights.

**Experiment setups.** We utilize Isaac-Gym [27] for parallel training. In the real-world experiment, we initialize ob-

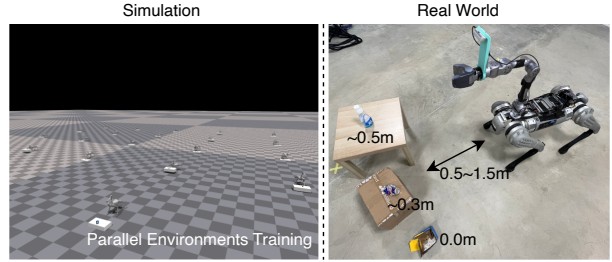

Figure 5: **Illustration of simulation / real-world setup.**

jects in a random location and orientation in front of the robot, making sure it is visible in at least one camera (around 0.5∼1.5m). We test three heights by putting the objects on the ground, on a box, and on a table, corresponding to the heights of 0.0m, ∼0.3m, ∼0.5m.

**Objects.** We use 33 YCB objects [28] in simulation. According to the object shapes, we roughly divided them into 7 representative categories: *ball*, *long box*, *square box*, *bottle*, *cup*, *bowl* and *drill*. As for real-world experiments, we choose 14 objects, including 4 irregular objects, 6 common objects, and 4 regular shapes. It is worth noting that almost all objects are unseen in the simulation. See Appendix H for illustrated details.

### 4.1 Simulation Results

**Evaluation principles and baselines.** In the simulation, we consider a successful grasp if the object is picked up and a failure when the object falls down the table or is not successful in 150 high-level steps ($\sim 18 - 24s$). We compare both the privileged teacher policies of VBC against baselines and their visuomotor student policies to provide an analysis of each module of VBC: 1) VBC w.o. shape feature: a visuomotor policy that is trained the same as VBC, but does not access the pre-trained object shape features during the teacher training stage. 2) Floating base: a floating base policy with a perfect low-level navigation ability but without arm-leg coordination; in other words, the robot can always follow the given velocity commands and body height commands ranging from 0.4 to 0.55 meters (we set 0.4 meters as the lower bound as the default controller in the real-world only allows 0.47 meters in minimum). We compare both the teacher and its

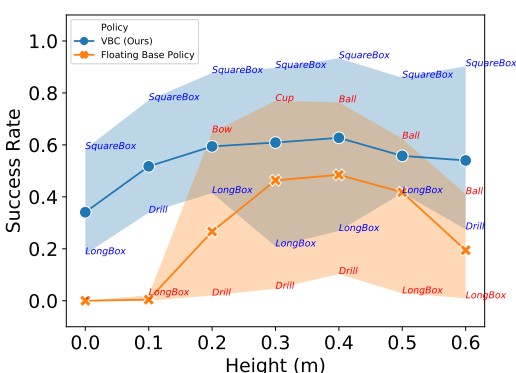

Figure 6: **Success rates** of different student policies at different heights in *simulation*. We test each distinct object for more than 100 episodes. The dots/crosses represent the *mean* performance, and the upper/lower bound is the max/min success rate over 7 categories, with the category name noted. The non-hierarchical baseline is not included for not reaching meaningful results.

corresponding student policy using the same policy network structure. 3) Non-hierarchical: a uni-

Table 1: **Success rates** of VBC compared to baseline method on seven categories of objects, tested in the *simulator*. Each distinct object is tested for more than 300 episodes over 3 random seeds. The robot starts from a 1.0-meter distance from the table. Every episode ends when the robot successfully picks up the object, the object falls from the table, or the robot fails until 150 steps. With the 3D shape feature, the algorithm is better at handling irregular objects.

| | Policy Type | Ball | Long Box | Square Box | Bottle | Cup | Bowl | Drill |
|---|---|---|---|---|---|---|---|---|
| **Privleged Teacher** | **Floating Base** | $0.63 \pm 0.01$ | $0.20 \pm 0.03$ | $0.73 \pm 0.01$ | $0.36 \pm 0.02$ | $0.71 \pm 0.02$ | $0.71 \pm 0.01$ | $0.07 \pm 0.02$ |
| | **VBC w.o. Shape Feature** | $\mathbf{0.96 \pm 0.01}$ | $0.74 \pm 0.03$ | $\mathbf{0.96 \pm 0.00}$ | $\mathbf{0.82 \pm 0.01}$ | $0.84 \pm 0.01$ | $0.55 \pm 0.01$ | $0.74 \pm 0.00$ |
| | **VBC (Ours)** | $0.89 \pm 0.01$ | $\mathbf{0.93 \pm 0.00}$ | $81.87 \pm 0.02$ | $0.76 \pm 0.01$ | $\mathbf{0.89 \pm 0.02}$ | $\mathbf{0.78 \pm 0.03}$ | $\mathbf{0.84 \pm 0.00}$ |
| **Visuomotor Student** | **Non-Hierarchical** | 0.0 | 0.0 | 0.0 | 0.0 | 0.0 | 0.0 | 0.0 |
| | **Floating Base** | $0.39 \pm 0.07$ | $0.06 \pm 0.02$ | $0.42 \pm 0.02$ | $0.15 \pm 0.03$ | $0.38 \pm 0.05$ | $0.44 \pm 0.01$ | $0.03 \pm 0.01$ |
| | **VBC (Ours)** | $\mathbf{0.67 \pm 0.07}$ | $\mathbf{0.42 \pm 0.12}$ | $\mathbf{0.74 \pm 0.09}$ | $\mathbf{0.55 \pm 0.01}$ | $\mathbf{0.74 \pm 0.06}$ | $\mathbf{0.52 \pm 0.02}$ | $\mathbf{0.47 \pm 0.11}$ |

(a) Objects on the ground (0.0m). (b) Objects on a box ($\sim$0.3m). (c) Objects on a table ($\sim$0.5m)

Figure 7: **Average success times w.r.t. the attempts** of picking up 14 objects at different heights, tested in the *real world*. We allow 5 continuing attempts (including 4 continuous retries, x-axis) during one configuration, i.e., one reset of picking up one specific object at a specific height. A failure is recorded if the robot attempts more than 5 attempts in one reset or if the object falls from its surface. We test 5 times for each object and each height, and calculate the average success time (y-axis) across these trials. *The emergent retrying behavior improves the performance.*

fied policy trained with low-level policy and visuomotor high-level policy jointly in an end-to-end style. This policy takes the observation of our low-level and high-level policies together and outputs all the target positions of 12 robot joint angles and the target pose of the gripper. We failed in training such a policy, indicating the effectiveness and necessity of VBC's training pipeline.

**Picking up different objects.** We test the picking-up *success rate* on each distinct object for more than 300 episodes and collect the results over every category. As is shown in Tab. 1, VBC achieves the best performance on 4/7 categories of objects. It is also interesting that with 3D features, VBC becomes worse than *VBC w.o. Shape Feature* at regular objects, but is much better on irregular objects with complicated shapes, i.e., *long box*, *cup*, *bowl*, and *drill*. One possible reason is that, without pre-trained features, the policy targets the object's center for grasping, which is effective enough on simpler objects yet inapplicable for objects with more complicated shapes. One may notice that the student policy is much worse than the teacher policy, since when the robot touches but fails to grasp on the first try, the object may be randomly laid so that its pose will be much harder to grasp within only 150 steps using only visual inputs. In addition, the performance of the student policy of VBC behaves the worst on *long boxes*. This is because without shape features, it is much harder to determine the pose of the object, and some initial poses of *long boxes* are extremely hard to grasp for the gripper we used at some height (also as shown in Fig. 6).

**Picking up over various heights.** We test the mean *success rate* on each object category on a fixed height, where we also list the object with the maximum/minimum success rate on each height. We compare VBC and the floating base baseline on 7 different heights, and illustrate the results in Fig. 6. From the significant improvement of VBC on almost every height, compared to the floating base policy, we highlight the advantage of VBC in achieving flexible whole-body behaviors.

## 4.2 Real-World Experiments

**Evaluation principles and baselines.** We deploy the trained visuomotor student policy of VBC directly into the real world. Particularly, we test how many times the robot can pick up the assigned object in five attempts without resetting (i.e., the robot is allowed to continuously retry four attempts when it fails to grasp), since we find that the emergent retrying behaviors improve the robustness, reliability, and success rate of finishing a job. We consider it a success when the object is picked more than 0.1m higher than the placed plane and a failure when the robot tries more than five

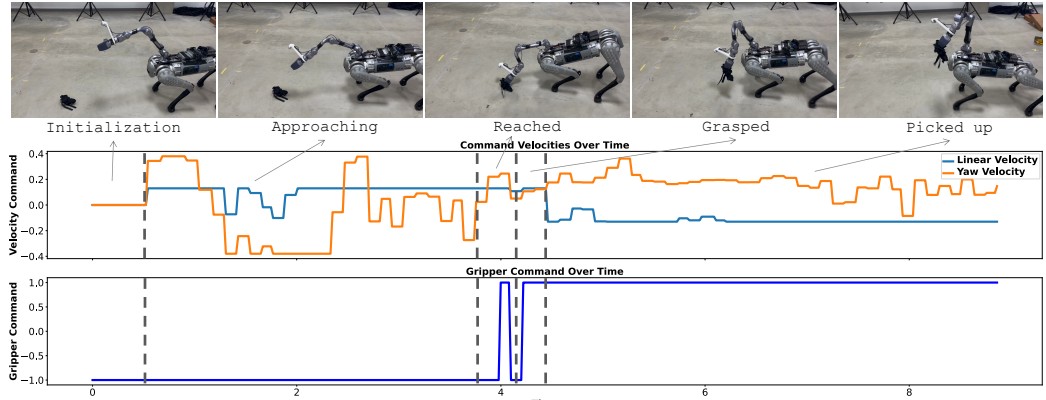

Figure 8: **Visualization of robot behavior and corresponding commands during the test time.** We plot the velocity commands (including the quadruped linear velocity and yaw velocity); along with the gripper command (close or open).

attempts or the object falls from its surface. Each object is evaluated over three different heights and we do five resets for each configuration. Regarding baselines, as it is hard to build a fair *autonomous* method for comparison, we compared VBC with teleoperating the default Unitree base controller (allowing the heights of the arm base to vary from 0.47m to 0.55m) combined with a similar high-level policy trained on stationary legs, in which the robot has no arm-leg coordination but also retains the retrying. For this baseline, We teleoperated the robot to be close enough to the object before grasping it. During all tests, we put objects at a distance that one mounted camera can observe, thus they can be easily manually annotated using the *TrackingSAM* tool. After the annotation, the robot walks towards and picks up the objects in a fully *autonomous* way. In experiments, we observe that the policy is capable of recovering from short-term tracking loss, which is not uncommon due to the robot turning or tilting.

**Emergent retrying behaviors.** The robot emerges to learn a retrying behavior in simulation training. In other words, when the robot fails to grasp an object, it automatically retries to grasp without human intervention. This shows an advantage and a clear difference compared to model-based methods. In our observation, as long as the visual mask is on-track/not-lost, the success rates benefit from 1) the change of the object pose in the early tries; and 2) the movement of the robot body.

**Pickup performance.** We collect all results and conclude the averaged performance over all objects on each height setting and show in Fig. 7. It is worth noting that our method allows for autonomous retrying behaviors when one pick-up trial fails. Obviously, VBC surpasses the baseline method on all settings. To highlight, the baseline methods fail at 0.0m and 0.3m, as the default controller keeps a fixed robot height, similar to the floating base baseline. Even on the 0.5m setting, where the teleoperated baseline without whole-body behavior is equivalent to a static robotics arm, VBC is generally better. Besides, VBC shows great generalization ability on unseen shapes, thanks to the training strategy and observation design in the training pipeline.

**Qualititive analysis.** We analyze whether the commands sent by the high-level policy are reasonable to call the low-level policy to achieve the task. To this end, we visualize the velocity commands, the gripper command, along with their corresponding robot behavior during the testing time in Fig. 8. Our system has small gaps when deploying our policy trained in the simulator into the real world, which is able to provide the correct commands that show clear phases.

## 5 Conclusions and Limitations

We proposed Visual Whole-Body Control (VBC), a fully autonomous loco-manipulation system on quadruped robots with a hierarchical training pipeline. VBC trains a low-level goal-reaching policy and a high-level task-planning policy through reinforcement learning and imitation learning.

**Limitations.** Although we showed great success in picking various objects, there are several limitations regarding the system and the tasks that can be of interest in future work. A detailed analysis is in Appendix I and we outline some reasons: (1) visual tracking loss; (2) inadequate gripper design; (3) in-precise depth estimation; (4) compounding error.

**Acknowledgments**

This project was supported, in part, by the Amazon Research Award, the Intel Rising Star Faculty Award, and gifts from Qualcomm, Covariant, and Meta. We thank Chengzhe Jia for the great help on 3D modeling. We also thank Haichuan Che, Jun Wang, Shiqi Yang, and Jialong Li for their kind help in recording demos. We thank Xuanbin Peng for helping with the experiments.

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

## A  Extended Related Works

**Legged Locomotion.** For decades, legged locomotion has been an essential topic in the field of robotics and has gained much success. Classical methods solve the legged locomotion task through model-based control to define controllers in advance. For example, Miura and Shimoyama [29], Sreenath et al. [30] utilized traditional methods to enable the walking of biped robots. MIT cheetah 3 [31], ANYmal-a [32] by ETH, and Atlas [33] by Boston Dynamics are good examples of robots designed with a robust walking controller. Based on these developed robots, researchers have been able to develop advanced controllers for even more agile locomotion tasks, such as jumping or walking over varying obstacles [34, 35, 36, 37, 38, 39]. To make a robust controller that can adapt to varying environments, researchers integrate some sensor data such as elevation maps [40, 41, 42, 43, 44] and state estimators [45, 46, 47, 48, 49]. Although classical methods have greatly succeeded in legged locomotion, generalizability in pre-designed controllers would require careful and tedious engineering efforts. With the fast development of deep neural networks, learning-based end-to-end methods have become a simple way to gain robust controllers with strong generalization abilities. Learning-based methods often train a robust controller in physics simulators such as PyBullet [50] and Isaac Gym [27] and then perform sim-to-real transfer to a real robot. For example, Rudin et al. [51] uses Isaac Gym with GPU to enable a quadruped robot to walk in only several minutes, Kumar et al. [52, 53] proposed the RMA algorithm to make the sim-to-real transfer process easier, and more robust, Lee et al. [54], Miki et al. [55], Agarwal et al. [4], Yang et al. [56], Yu et al. [57], Cheng et al. [3], Zhuang et al. [5], Fu et al. [58], Margolis and Agrawal [59], Li et al. [60] enable the legged robots to walk or run on varying terrains, and Fuchioka et al. [61], Escontrela et al. [62], Peng et al. [63], Yang et al. [64], Wang et al. [65], Li et al. [66], Kumar et al. [67] combine RL with imitation learning to produce naturalistic and robust controllers for legged robots. Kim et al. [68] map human motions to a quadruped robot to complete a wide range of locomotion and manipulation tasks.

**Mobile Manipulation.** Previously mentioned works mainly focus only on the mobility part; adding manipulation to the mobile base is yet to be well-solved. Autonomous mobile manipulation on wheeled robot platforms has made significant progress over the past few years. Wong et al. [69] collected teleoperated mobile manipulation data and learned imitation-learning-based policies in simulation. Lew et al. [70] built an efficient framework for table cleaning with RL and trajectory optimization. Garrett et al. [71] and Srivastava et al. [72] integrate task-specific planner for various mobile manipulation tasks. Gu et al. [73] and Xia et al. [74] developed modularized mobile manipulators with RL, Yokoyama et al. [75] introduced a high-level controller for coordinating different low-level skills. Sun et al. [76] designed a modularized policy with components for manipulation and navigation, trained by RL to learn navigation and grasping indoors. Yang et al. [77] proposed simultaneously controlling the robot arm and base for harmonic mobile manipulation using RL training with large-scale environments. Ahn et al. [78] pre-trained a set of single mobile or manipulation tasks and relied on large language models to decompose and call the skills to achieve long-horizon tasks. Brohan et al. [79] train a robotics transformer to perform real-world tasks given human instructions, trained on large-scale and real-world demonstration. Shafiullah et al. [80] learned to solve household tasks from 5-minute demonstrations based on pre-trained home representations. While encouraging, most of these mobile manipulators with wheels cannot operate with challenging terrains in the wild.

## B  Notations and Background

### B.1  Reinforcement Learning

Reinforcement learning (RL) is usually formulated as a $\gamma$-discounted infinite horizon Markov decision process (MDP) $\mathcal{M} = \langle \mathcal{S}, \mathcal{A}, \mathcal{T}, \rho_0, r, \gamma \rangle$, where $\mathcal{S}$ is the state space, $\mathcal{A}$ is the action space, $\mathcal{T} : \mathcal{S} \times \mathcal{A} \to \mathcal{S}$ is the environment dynamics function, $\rho_0 : \mathcal{S} \to [0,1]$ is the initial state distribution, $r(s,a) : \mathcal{S} \times \mathcal{A} \to \mathbb{R}$ is the reward function, and $\gamma \in [0,1]$ is the discount factor. The goal of RL is to train an agent with policy $\pi(a|s) : \mathcal{S} \to \mathcal{A}$ to maximize the accumulated reward $R = \sum_t \gamma^t r(s_t, a_t)$.

Table 2: Details of low-level policy.

Definition of Symbols

| Name | Symbol |
|---|---|
| Leg joint positions | $\mathbf{q}$ |
| Leg joint velocities | $\dot{\mathbf{q}}$ |
| Leg joint accelerations | $\ddot{\mathbf{q}}$ |
| Target leg joint positions | $\mathbf{q}^*$ |
| Leg joint torques | $\tau$ |
| Base linear velocity | $v_b$ |
| Base angular velocity | $\omega_b$ |
| Base linear velocity command | $v_x^*$ |
| Base angular velocity command | $v_{\text{yaw}}^*$ |
| Number of collisions | $n_c$ |
| Feet contact force | $\mathbf{f}^{\text{foot}}$ |
| Feet velocity in Z axis | $\mathbf{v}_z^{\text{foot}}$ |
| Feet air time | $t_{\text{air}}$ |
| Timing offsets | $\theta^{\text{cmd}}$ |
| Stepping frequency | $f^{\text{cmd}}$ |

Reward Function Details for the Low-Level Policy

| Name | Definition | Weight |
|---|---|---|
| Linear velocity tracking | $\phi(v_{b,xy}^* - v_{b,xy})$ | 1 |
| Yaw velocity tracking | $\phi(v_{\text{yaw}}^* - \omega_b)$ | 0.5 |
| Angular velocity penalty | $-\|\omega_{b,xy}\|^2$ | 0.05 |
| Joint torques | $-\|\tau\|^2$ | 0.00002 |
| Action rate | $-\|\mathbf{q}^*\|^2$ | 0.25 |
| Collisions | $-n_{collision}$ | 0.001 |
| Feet air time | $\sum_{i=0}^{4}(t_{air,i} - 0.5)$ | 2 |
| Default Joint Position Error | $\exp\left(-0.05\|\mathbf{q} - \mathbf{q}_{\text{default}}\|\right)$ | 1 |
| Linear Velocity z | $\|v_{b,z}\|^2$ | $-1.5$ |
| Base Height | $\|h_b - h_{b,\text{target}}\|$ | $-5.0$ |
| Swing Phase Tracking (Force) | $\sum_{\text{foot}}\left(1 - C_{\text{foot}}^{\text{cmd}}(\theta^{\text{cmd}}, t)\right)\left(1 - \exp\left(-\|(\mathbf{f}^{\text{foot}})^2/\sigma_{cf}\|\right)\right)$ | $-0.2$ |
| Stance Phase Tracking (Velocity) | $\sum_{\text{foot}}\left(C_{\text{foot}}^{\text{cmd}}(\theta^{\text{cmd}}, t)\right)\left(1 - \exp\left(-\|(\mathbf{v}_z^{\text{foot}})^2/\sigma_{cv}\|\right)\right)$ | $-0.2$ |

## B.2 Proximal Policy Optimization

Proximal policy optimization (PPO) [81] is one of the popular algorithms that solve RL problems. The basic idea behind PPO is to maximize a surrogate objective that constrains the size of the policy update. In particular, PPO optimizes the following objective:

$$L^{PPO}(\theta_\pi) = \mathbb{E}_\pi[\min(rA, \text{clip}(r, 1 - \epsilon, 1 + \epsilon)A)], \tag{1}$$

where $r = \frac{\pi(a|s)}{\pi_{old}(a|s)}$ defines the probability ratio of the current policy and the old policy at the last optimization step, $A(s, a)$ is the advantage function, and $\theta_\pi$ is the parameter of the policy $\pi$.

## B.3 Imitation Learning

In general, imitation learning (IL) [82, 83] studies the task of learning from expert demonstrations. In this work, instead of learning from offline expert data, we refer to an online IL method, dataset aggregation (DAgger) [82] such that we request an online expert to provide the demonstrated action. Formally, the goal is to train a student policy $\hat{\pi}$ minimizing the action distance between the expert policy $\pi_E$ under its encountered states:

$$\hat{\pi} = \arg\min_{\pi \in \Pi} \mathbb{E}_{s \sim d_\pi}[\ell(s, \pi)]. \tag{2}$$

In this paper, $\ell$ is the mean square error in practice.

## B.4 Whole-Body Control

Whole-body control in robotics refers to controlling a mobile manipulator with the function of locomotion and manipulation by one unified framework [9, 8]. In this work, we consider a system that contains a quadruped robot combined with a 6-DoF robot arm, each of which has an underlying PD controller, and we use a learned policy to generate control signals and feed them to the controller given 1) the robot states and 2) commands.

In detail, the policy used for controlling the quadruped robot is driven by external commands, such as the base velocity command; and the arm is controlled by a PD controller driven by the $\mathbb{SE}(3)$ end-effector position-orientation command and inverse kinematics.

## C More Training Details

### C.1 Low-Level Training

**Reward functions.** Our reward for training the low-level policy for whole-body control mainly consists of four parts: command-following reward, energy reward, alive reward, and phase reward. The command following reward encourages the robot to track and follow the provided commands and explore various whole-body behaviors; the energy reward penalizes energy consumption to enable smooth and reasonable motion; the alive reward encourages the robot not to fail; the phase reward encourages the robot to walk in a steady behavior as mentioned above. The detailed definitions of

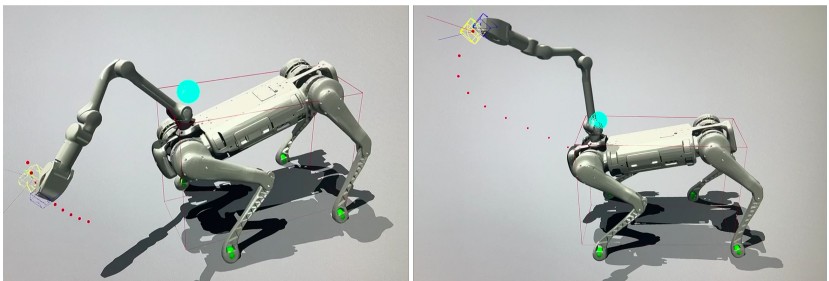

Figure 9: **Screenshots of low-level training**, where the robot is required to track the given velocity commands (linear and yaw) and the end-effector pose. We employ a height-invariant coordinate system originating at the robot's base, such that the sampled goal poses are only affected by the robot's horizontal movement (translation in X, Y axes and orientation in the yaw axis). The cyan point represents the origin of the resampled goals; the blue point and the yellow point are the current end-effector position and the goal position, respectively. The red dots are the interpolated goals. The red lines represent the space for determining self-collision when resampling new goals.

the low-level reward functions are shown in Tab. 2, where we define $\phi = \exp(-\frac{\|x\|^2}{0.25})$ for tracking the desired velocity of the robot. It is worth noting that we introduce the swing phase tracking rewards to help the robot learn trotting, and $C_{\text{foot}}^{\text{cmd}}(\theta^{\text{cmd}}, t)$ computes the desired contact state of each foot from the phase and timing variable, as described in Margolis and Agrawal [59].

**Low-level commands sampling.** To train a robust low-level controller for goal-reaching, we should sample and track various goals (i.e., base velocity commands and end-effector commands) as possible. As for the base velocity commands, we uniformly sample the linear velocity from a range of $[-0.6m/s, 0.6m/s]$, and the angular velocity from a range of $[1.0rad/s, 1.0rad/s]$.

Regarding the end-effector, same to Fu et al. [8], we employ a height-roll-pitch-invariant coordinate system originating at the robot's base, where the sampled goal poses are only affected by the robot's horizontal movement instead of changes in height. In detail, the origin (cyan point in Fig. 9) is set on a fixed plane, which is on the same line of Z axis as the arm base. Therefore, the sampled goals are also not affected by the robot's height, and orientation of the roll and pitch axis, but only by the orientation in the yaw axis. Such a design ensures the sampled trajectories are always on one local sphere at any time, guaranteeing the consistency of these samples. Besides, it helps to learn a whole-body behavior to control the legs and the body height to enable the arm to reach an expanded space of goals. If not, the robot may fail to reach a goal out of the workspace of the arm without bending or rising. Therefore, the body and the arm are not decoupled, although we use IK to control the arm.

We parameterize the end-effector position command $\mathbf{p}^{\text{cmd}}$ in spherical coordinate $(l, p, y)$, where $l, p, y$ lie in ranges of [0.4, 0.95], [-1 * $\pi$ / 2.5, 1 * $\pi$ / 3], [-1.2, 1.2], respectively. To make a smooth end-effector movement, we set $\mathbf{p}^{\text{cmd}}$ by interpolation between the current end-effector position $\mathbf{p}$ and a randomly sampled end-effector position $\mathbf{p}^{\text{end}}$, at intervals of $T_{\text{traj}}$ seconds:

$$\mathbf{p}_t^{\text{cmd}} = \frac{t}{T_{\text{traj}}}\mathbf{p} + \left(1 - \frac{t}{T_{\text{traj}}}\right)\mathbf{p}^{\text{end}}, t \in [0, T_{\text{traj}}].$$

$\mathbf{p}^{\text{end}}$ is resampled in a fixed timestep or if any $\mathbf{p}_t^{\text{cmd}}$ leads to self-collision or collision with the ground. Fig. 9 shows screenshots of the low-level training, where the robot is required to track the given velocity commands and the end-effector pose. The orientation command $\mathbf{o}^{\text{cmd}}$ is uniformly sampled from $\mathbb{SO}(3)$ space.

**Regularized online adaptation for sim-to-real transfer.** We take the training paradigm of Regularized Online Adaptation (ROA) [8, 84, 17] in the RL training, to help decrease the realizability gap when deploying our system into the real world. As illustrated in Fig. 10, in the first training phase, we train an encoder $\mu$ that takes the privileged information $e$ (such as friction and mass...etc.) as inputs, and predicts an environment extrinsic latent vector $z^\mu$, which will be used for policy inference. Note that the privileged information is not available in real robot deployment. Thus we will need another way to estimate the latent vector $z^\mu$. To this end, we train the adaptation module $\phi$ in a

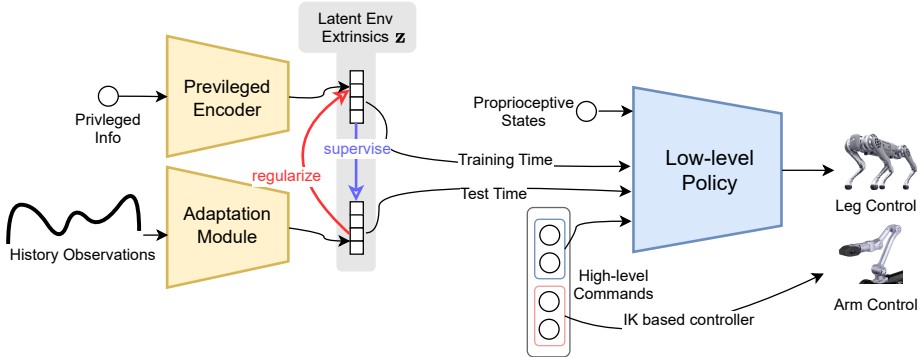

Figure 10: **Regularized Online Adaptation (ROA) training of the low-level policy.**

second training phase, to imitate/distill the output of the encoder $\mu$, and estimate this $z^\mu$ only based on recent history low-level observations. The loss function of the low-level policy w.r.t. policy's parameters $\theta_{\pi^{\text{low}}}$, privileged information encoder's parameters $\theta_\mu$, and adaptation module's parameters $\theta_\phi$, can be then formulated as:

$$L\left(\theta_{\pi^{\text{low}}}, \theta_\mu, \theta_\phi\right) = -L^{PPO}(\theta_{\pi^{\text{low}}}, \theta_\mu) + \lambda \left\| z^\mu - \text{sg}\left[z^\phi\right] \right\|_2 + \left\| \text{sg}\left[z^\mu\right] - z^\phi \right\|_2 \quad (3)$$

where $\text{sg}\left[\cdot\right]$ is the gradient stop operator, and $\lambda$ is the Laguagrian multiplier. It is worth noticing that RMA [52] is a special case of ROA, where the Laguagrian multiplier $\lambda$ is set to be constant zero. The adaptation module $\phi$ starts training only after convergence of the policy $\pi^{\text{low}}$ and the encoder $\mu$.

**Domain randomization.** When training, we randomize the terrains, including flat planes and rough ones. We also randomized the friction between robots and terrains. Besides, we randomized the quadruped's mass and center of mass. These randomizations, together with our ROA module, contribute to a robust low-level policy.

**Policy architecture.** The low-level control policy network is a three-layer MLP, where each layer has 128 dimensions and ELU activations. It receives observations, including the extrinsic latent environment, which is encoded by the privileged encoder during training, and the adaptation module during testing. The privileged information encoder that models environment physics parameters is a two-layer MLP with a hidden size of 64; the adaptation module encodes history rollouts, which is a two-layer convolutional network.

### C.2 High-Level Training

**Reward functions.** The task-specific reward function for training our state-based policy comprises stage rewards and assistant rewards. 1) We design three *stages* for a task. The first stage is approaching, and the corresponding reward $r_{\text{approach}}$ encourages the gripper and quadruped to get close to the target object: $r_{\text{approach}} = \min(d_{\text{closest}} - d, 0)$ ,where $d_{\text{closest}}$ is the current minimum distance between the gripper and the target object. The second stage is task progress, and the reward $r_{\text{progress}}$ leads the robot to the desired final state, e.g., lifting the object in the pickup task: $r_{\text{progress}}^{\text{pickup}} = \min(d - d_{\text{highest}}, 0)$ ,where $d_{\text{highest}}$ is the current largest height of the object. The third stage is denoted task completion, and reward $r_{\text{completion}}$ bonus when the agent completes the task. $r_{\text{completion}} = \mathbf{1}(\text{Task is completed})$.For example, in the pickup task, the task completion condition is that the object height exceeds the required threshold. Note that at one time, the robot can only be at one stage and receive one stage reward of the above three, ensuring that different task stages are not intervened in. The assistant rewards are designed to a) smooth the robot's behavior to be easier to transfer to the real world, like penalty huge action commands and joint acceleration; b) prevent deviation and sampling useless data, such as acquiring the robot's body and gripper to point to the object, which guarantees that the camera does not lose tracking objects. 2) Beyond these stage rewards, we also design a set of *assistant* reward functions that help accelerate the learning procedure. We put all reward functions along with their weights of training the high-level policies

Table 3: **Details of high-level rewards**, where $\dot{\mathbf{q}}$ denotes arm joint velocities; $v_x^*$ denotes base velocity command; $\mathbf{d}_{\text{obj}}$, $\mathbf{d}_{\text{ee}}$ and $\mathbf{d}_{\text{base}}$ are the direction from robot body to the object, the direction from end-effector to the object and the direction of the robot body orientation, separately; $\mathbf{x}_{\text{obj}}$ and $\mathbf{x}_{\text{base}}$ are the position coordinates in world frame of the object and the robot body. $r_{\text{approach}}$, $r_{\text{progress}}$, and $r_{\text{completion}}$ are defined at Section 3.2.

|  | **Definition** | **Weights** |
|---|:---:|:---:|
| Stage Rewards | $r_{\text{approach}}$ | 0.5 |
|  | $r_{\text{progress}}$ | 1.0 |
|  | $r_{\text{completion}}$ | 3.5 |
| Assistant Rewards | $r_{\text{acc}} = 1 - \exp(-\|\dot{\mathbf{q}}_{t-1} - \dot{\mathbf{q}}_t\|)$ | $-0.001$ |
|  | $r_{\text{cmd}} = -\|v_x^* + 0.25\exp(-\|v_x^*\|)$ | 1.0 |
|  | $r_{\text{action}} = 1 - \exp(-\|\mathbf{a}_{t-1} - \mathbf{a}_t\|)$ | $-0.001$ |
|  | $r_{\text{ee orn}} = \cos(\mathbf{d}_{\text{obj}} \cdot \mathbf{d}_{\text{ee}})$ | 0.01 |
|  | $r_{\text{base orn}} = \cos(\mathbf{d}_{\text{obj}}, \mathbf{d}_{\text{base}})|$ | 0.25 |
|  | $r_{\text{base approach}} = (1 + \tanh(-10 * \|x_{obj} - x_{base} - 0.6\|))$ | 0.01 |

in Tab. 3. Except for $r_{\text{progress}}$, $r_{\text{completion}}$ and $r_{\text{completion}}$ that is determined by the task stage, we define a set of assistant rewards to help the algorithm converge faster. Among them, $r_{\text{acc}}$ limits the arm joints velocities' change rate; $r_{\text{cmd}}$ is designed to encourage the robot to slow its velocity when it is close to the object to be picked., $r_{\text{action}}$ helps smooth the output of the high-level policy to prevent it from changing quickly; $r_{\text{ee orn}}$ and $r_{\text{base orn}}$ encourages the arm and the robot body to be aimed at the object; $r_{\text{base approach}}$ encourages the robot to be close to the object when the robot is far away from the object.

**Teacher policy network architecture.** Our privileged teacher high-level policy network includes a pre-trained PointNet encoder for modeling object pointclouds, followed by a two-layer MLP that gives the high-level actions.

**Student policy network architecture.** Our vision-based high-level policy network comprises a CNN and a two-layer MLP that outputs the high-level actions. We concatenate the images and other states as observations. We stack 4-step images which are passed through the two-layer CNN network with kernel size being 5 and 3, respectively, and then encoded as a 64-dim latent vector. Then, we concatenate this latent vector with other states as a flattened vector. This vector is then passed to a two-layer MLP with a hidden size of 64. We use ELU as the activation.

**Simulation** We use the GPU-based Isaac Gym simulator [27]. As for the teacher, we train 10240 environments in parallel; regarding the student, we only utilize 240 parallel environments as there is a huge GPU memory cost for rendering visual observations. Experiments are trained on a single GPU (we use GTX4090 and GTX3090). On a GTX4090, The teacher policy takes around 36 hours for training, and the student policy consumes around 48 hours.

**Additional training techniques.** We design several training techniques with our prior knowledge to help the high-level policy to be well-performed and stable, not only in the simulation, but can also be transferred into the real world. We briefly introduce them in the following:

1. Action delay. We add a one-step action delay to cover the inference and execution in the real world.

2. Command clip curriculum. We gradually clip the linear velocity command during training, as we find that a large velocity helps accelerate the learning convergence but a slow velocity ensures a safe, stable, and smooth behavior in the real world.

3. Reward curriculum. The command penalty reward $r_{\text{cmd}}$ is added after the robot learns well to pick up objects.

4. Randomly changing the object position. During training, we take a small probability (10%) to randomly change the object position and pose. This helps the robot to learn the retrying behaviors when it fails during one trial.

5. Forcing stop when closing gripper. We forcibly set the velocity command to 0 when the gripper command is closing. This contributes a lot to a stop-then-pick behavior and thus improves the performance.

These training techniques are adopted for both teacher policy and student policy training. When training the student policy, we additionally randomize the camera latency as the policy will never obtain the immediate images at the time it infers, but some pictures that are a few milliseconds before, similar to the real world. We randomize the value by measuring the camera latency of the device we used in our real-world experiments.

## D   Summary of Domain Randomization

We summarize and list all domain randomization here during the simulated training. **Low-level policy.** The randomization of the low-level policy is to train a robot with robust walking skills that adapt to different surroundings, while also knowing how to adjust body poses to reach different end-effector goals.

- Terrains, including flat planes and rough ones.
- Friction between robots and terrains.
- Mass of the quadruped and the arm.
- Center of mass of the robot.
- Motor strength.

**High-level teacher policy.** The randomization of the high-level part mainly focuses on learning the manipulation task with generalization. The high-level teacher policy follows the same randomization as the low-level policy, along with the following additional terms:

- High-level frequency by involving random low-level policy calls during one high-level call. In particular, we randomize 6-8 calls considering the high-level inference time and signal transmission latency, corresponding to 0.12-0.16s (low-level policy run at 50 Hz), i.e., 6.25-8.33Hz.
- Kp and Kd parameters for the arm controller.
- Table heights, 0-0.5m.
- Initial position and pose of the robot.
- Position and pose of objects.

**High-level student policy.** The high-level student policy follows the same randomization as the teacher policy, along with the following additional term:

- Camera latency.

## E   Motivations for IK-based Arm Control

In this work, we control the arm by solving the target joints with inverse kinematics, given the predicted target end-effector 6-DoF poses, rather than directly predicting the arm joints. Such a design is motivated by lessons learned from the previous work, DeepWBC [8], which struggles to control the accurate end-effector position and orientation. To verify the advantage of our IK-based control predicting end-effector poses compared to predicting the target joints, we conduct a goal-reaching experiment, in which we randomly sample end-effector goals as in training, and end the episode once the goal is reached (the distance of end-effector and the goal is less than 2cm) or the episode time is out. We quantitatively evaluate the average error of position and orientation in Tab. 4. The results demonstrated a clearly better precision compared to the target joint prediction solution in Fu et al. [8], which contributes a lot to training an effective high-level policy to complete complicated manipulation tasks.

## F   Sim-Real Visual Observation Comparison

We compare the trajectories and segmented depth images in simulation and real-world sampled by VBC, shown in Fig. 11.

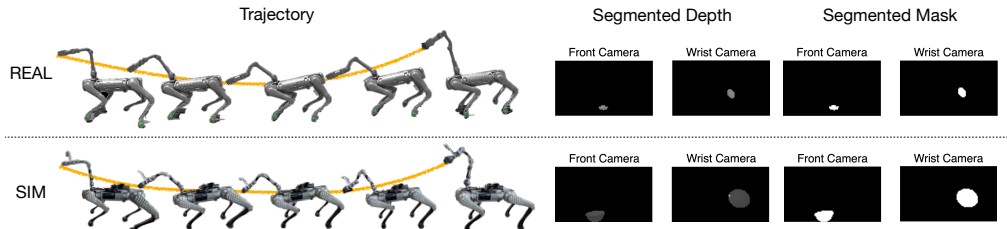

Figure 11: **Comparison between trajectories and segmented depth images in simulation and real-world sampled by VBC**. Left: The real-world behaviors match well with the simulation, indicating a less sim-to-real gap in our method. Right: Our TrackingSAM-based working pipeline allows us to obtain the segmentation mask precisely during deployment (not the same object observed). The depth images are pre-processed by hole-filling and clipping.

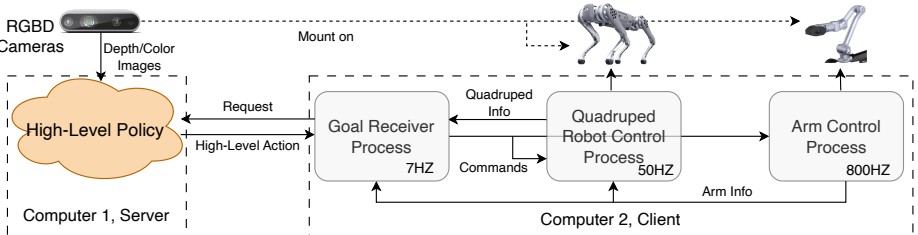

Figure 12: **The server-client multi-processing system design.** Modules work asynchronously under different frequencies. The high-level planning policy (left) and the low-level control modules (right, controlling arm and legs) are deployed on different devices. To be more specific, the goal receiver requests the high-level policy to take the visual input and receive the new commands; the low-level policy sends joint position commands to the quadruped runs at 50Hz, given the input of high-level commands; the IK-based arm controller runs and send the joint position command to the arm constantly at 800Hz, given the target end-effector pose. Therefore, the low-level policy and the arm controller keep utilizing the old commands to control the robot until the high-level policy updates the commands (i.e., target robot velocities and end-effector pose).

## G   Real World System

**Hardware setup.**   As visualized in Fig. 4, we mount two cameras on our robot platform, which are both RealSense D435 cameras. B1 itself has an onboard computer, and we also attached an additional onboard computer, Nvidia Jetson Orin, to the robot. The B1, Z1, and the Orin computer are all powered by B1's onboard battery. The inference of the low-level policy and the control of the arm is conducted at the onboard computer of B1, and that of the high-level vision-based planning policy is made on Nvidia Jetson Orin. It is worth noting that the computation and the power of our robot system are fully onboard and, therefore, untethered by any PCs.

**Software system design.**   During deployment, the low-level policy that controls the quadruped works at a frequency of 50Hz; in the meantime, the high-level policy runs at ∼8Hz, which is more than five times slower than the low-level policy; besides, the arm is required to work at a frequency

| Approach | Status | Position Error ($m$) ↓ | | | Orientation Error ($rad$) ↓ | | |
|---|---|---|---|---|---|---|---|
| | | Mean | Min | Max | Mean | Min | Max |
| Joint prediction ([8]) | Standing | $0.236 \pm 0.0$ | $0.146 \pm 0.0$ | $0.325 \pm 0.0$ | $1.154 \pm 0.0$ | $0.616 \pm 0.0$ | $1.686 \pm 0.0$ |
| EE prediction+IK (ours) | | $\mathbf{0.047 \pm 0.0}$ | $\mathbf{0.027 \pm 0.0}$ | $\mathbf{0.067 \pm 0.0}$ | $\mathbf{0.850 \pm 0.01}$ | $\mathbf{0.160 \pm 0.0}$ | $\mathbf{2.269 \pm 0.05}$ |
| Joint prediction ([8]) | Walking | $0.414 \pm 0.0$ | $0.182 \pm 0.0$ | $0.558 \pm 0.0$ | $2.015 \pm 0.0$ | $1.165 \pm 0.03$ | $3.198 \pm 0.04$ |
| EE prediction+IK (ours) | | $\mathbf{0.029 \pm 0.0}$ | $\mathbf{0.008 \pm 0.0}$ | $\mathbf{0.050 \pm 0.0}$ | $\mathbf{0.332 \pm 0.02}$ | $\mathbf{0.071 \pm 0.03}$ | $\mathbf{0.743 \pm 0.08}$ |

Table 4: Precision of reaching goals, evaluated in simulation with 3 seeds, each over 100 episodes. We record the results while the robots are standing and walking. In both status our solution achieves the best performance.

| | High-Level | | | | | Low-Level |
|---|---|---|---|---|---|---|
| | TrackingSAM | | | Pre-Process | Model Inference | Model Inference |
| | SAM clicking | AOT Init | AOT Tracking | | | |
| Time (s) | $1.369 \pm 0.33$ | $0.323 \pm 0.72$ | $0.083 \pm 0.0$ | $0.031 \pm 0.01$ | $0.003 \pm 0.0$ | $0.011 \pm 0.0$ |
| Stage | Initialization | | | Autonumous | | |
| Device | External NVIDIA Jetson Orin 64GB | | | | | Internal NVIDIA Jetson Xavier NX |

Table 5: **Component runtime analysis.** All components are run on real robots, and the results are averaged over 100 times.

of more than 800Hz. Since these policies work on different computers, along with the quadruped and the arm should work under different frequencies, we design a client-server multi-processing system for this work, as shown in Fig. 12. We set up three processes at the low-level policy side: a quadruped robot control process, an arm control process, and a goal receiver process behaving as a client. The high-level policy combines the vision inputs from the depth cameras and the information from the low-level request as the observation works as a server. Every 0.1 seconds, the goal receiver sends a request with the necessary information to the server; the high-level policy makes the latest inference and sends the high-level action back to the robot; at the same time, the quadruped and the arm control processes gather the latest high-level commands to execute, while keep updating the necessary information for the goal receiver process.

**Component runtime analysis.** We record the runtime of all our components running in real-world experiments, shown in Tab. 5. Before an autonomous working pipeline, users are required to click on images to annotate the object to be grasped with SAM, which is further processed by AOT to initialize the first reference frame to track. After initialization, the high-level process reads in a new frame from two cameras, and pre-process them by resizing, hole-filling, and clipping; the images will be further processed by TrackingSAM to track and provide segmentation masks; the masks and segmented images are concatenated as observations along with the proprioception states sent by the low-level process, and call the high-level model to predict the command. The high-level process finally sends the command back to the low-level process, which calls the low-level model to predict the leg joints. Note that we did not record the time cost of process communication between the high-level and the low-level, each of which runs on different devices and communicates through HTTP. During training, we gave a slightly larger randomization range than the recorded total time (also see in Appendix D). Overall, the high-level runtime limits the control frequency of the high-level policy to 6-8Hz. We also fix the low-level control frequency as 50Hz, even if the policy inference is rather fast.

## H    Details of Used Objects

For the simulation, we included 34 objects adapted from the YCB dataset [28]. According to the object shapes, we roughly divided them into 7 categories: *ball*, *long box*, *square box*, *bottle*, *cup*, *bowl* and *drill*. In the real-world experiments, we choose 14 objects, including four irregular objects, six common objects, and four regular shapes. Objects are illustrated in Fig. 13

## I    Failure Analysis and Future Works

Although we showed great success in picking various objects, we found several failure cases due to the limitations of the system: 1) Compounding error. Our pipeline is hierarchical and requires each module to be coherent and precise to make the robot work well. 2) In-precise depth estimation. The RGBD camera does not reflect a perfect depth estimation, especially on reflective objects. 3) Bad gripper. The gripper of the Unitree Z1 Arm is a beak-like gripper instead of a parallel one, which tends to push objects away and is hard for accurate manipulation. 4) Unstable vision inputs. The tracking SAM that is used to provide the mask of annotated objects, may lose track due to camera deviation and occlusion, or be confused when the selected objects have a similar color to the other things. This is the most common cause leading to failure in our real-world experiments. In our current pipeline, before the robot autonomously picks up objects, we must manually specify the target object in both cameras mounted on the robot. Thanks to the generalizability of trackingSAM, we

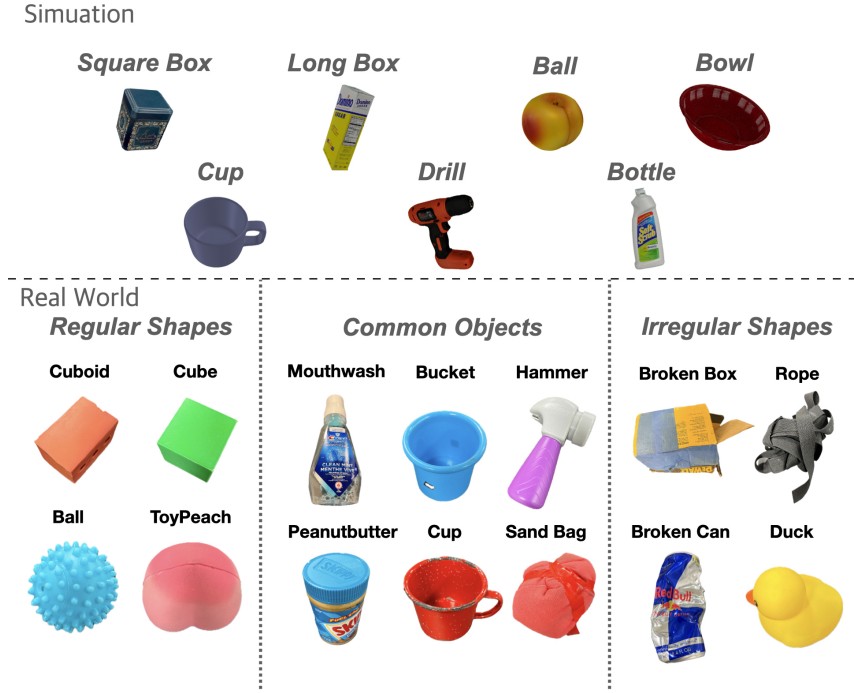

Figure 13: **Examples of simulated training objects and real-world objects.**

find that as long as the object is visible to one camera (either the head or the wrist camera), the annotation mask can be propagated to another view for automated picking, and free the robot from being specified in the between of the range of the two cameras, which improves the usability of our system. Since the requirements of annotation and object visibility are totally decoupled from our sim-to-real framework, in the future we can further improve and replace these modules with automated annotation algorithms and a search/exploration module to further improve the automation of the whole system. Our choice of visual inputs—specifically masks and segmented depth images—deliberately excludes RGB images that provide color information. This limitation restricts the system from performing tasks that rely on color differentiation, such as distinguishing between Fanta and Sprite cans, sorting garbage by category, or pressing a green button. During the training of our high-level policy, we utilized 33 distinct objects. While this approach demonstrates strong generalization capabilities on unseen objects in real-world scenarios, future work could enhance this by introducing randomized object sizes to further improve generalization. Future works should also consider upgrading the hardware, like replacing a better gripper and RGBD cameras that provide better depth estimation; and simplify the working pipeline.

## J Additional Details of TrackingSAM

TrackingSAM [24] provides a very simple and intuitive way to provide segmentation masks which only requires minimal manual intervention. It invokes SAM [26] to generate an initial segmentation mask of objects before the tracking begins, where the user only needs to make a few clicks (as shown in Fig. 14). Given the initial segmentation, we use it to populate two instances of trackers for images from the wrist camera and the head camera, respectively. The tracker used is AOT (Associating Objects with Transformers) [25], using the most lightweight AOT-Tiny variant, which is implemented based on MobileNet and can achieve nearly 10 fps on the onboard computer.

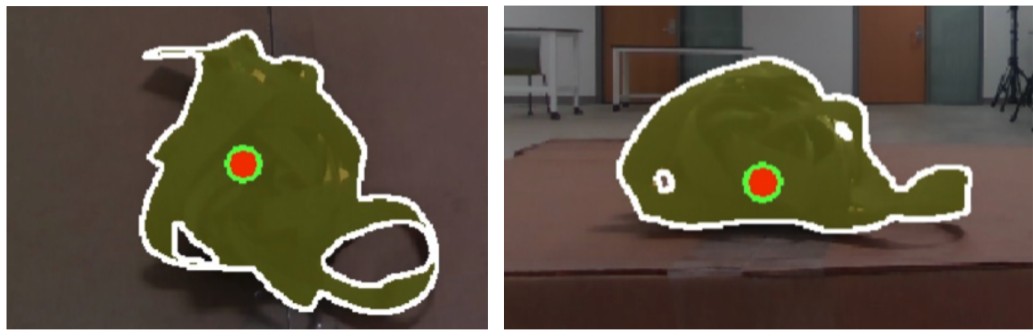

Figure 14: **Segmented examples of the TrackingSAM tool**.

