# OpenReview forum: "Visual Whole-Body Control for Legged Loco-Manipulation"
_robot-learning.org/CoRL/2024/Conference — CoRL 2024_

### Official Review · Reviewer_xf7y · 2024-07-19
**Review of Submission Number 12**

**Originality:** 4
**Technical Quality:** 4
**Clarity Of Presentation:** 4
**Potential Impact:** 4
**Recommendation:** 4
**Confidence:** 5

**Review:**

This paper tackles a very relevant problem for a quite interesting application. The problem of whole-body loco-manipulation has been quite extensively explored in the related literature, even for tasks which are more difficult than those shown in this paper, e.g., door opening. However, extending such frameworks with autonomous mobile manipulation capabilities which exploit vision is a very unexplored topic. I also find it impressive that the authors are able to learn policies from simulation without any human demonstrations. In addition, the paper has a very large number of pick-and-place experiments in many different scenarios, which show the generality of the approach with respect to many elements, such as object types and shapes, different terrains, and object heights.
However, I would appreciate if the authors could provide more details about:
- the model which is used to annotate objects and output segmentation masks;
- the computation time of each module.
The readability of the paper can be improved - there are many long paragraphs.

**Quality Of The Limitations Section:**

3

**Questions For Rebuttal:**

- Section 3.1: "any given end-effector pose and body velocity".
- Section 3.1: It is not clear in which frame v_{lin}^{cmd} and omega_{lin}^{cmd} are expressed. Are they expressed in base frame? Why are they scalar values and not vectors? Please improve the description here.
- Section 3.1 - observations: It is not clear what the variable z_t is. The paper says that it encodes environment physics, but it should be described better and more explicitly.
- Section 3.1 - Paragraph starting with "Besides, following the work of [4], etc...": please improve the readability.
- Do the "encoded shape features" mentioned in 3.2.1 coincide with the latent vector obtained by applying PointNet++ to the objects' pointclouds?
- Paragraph 3.2.1: it seems that the authors do not randomize the shape of the objects to pick up. Why? It seems that this would be an important variable to randomize.
- Section 3.3. Can the authors provide more details about the framework used to obtain segmentation masks in real-time ("TrackingSAM")? Is there a publication regarding this? [24] is a PhD thesis. Also, how reliable is this framework? Does it output segmentation masks which are 100% correct?
- Section 3.3: The authors state that all the computations happen on the robot on-board computers, which is quite impressive. Even though evaluating the networks used for the low-level and high-level policies is probably computationally feasible, I imagine that the model used to compute segmentation masks might be quite large. Can you comment about the computation time of all these modules?
- Section 4.1: a failure is considered if the object is not picked up in "150 high-level steps". What does this corresponds to in seconds?
- The authors state that the objects are manually annotated using trackingSAM. Can the authors specify why is this annotation needed and how is it performed? Does the user need to draw a bounding box around the object?
- Supplementary material: In some videos, the authors show the RGB image, masked depth, and segmentation mask from the two camera views. For the sake of clarity, I would suggest to show only the masked depth and segmentation mask, because the RGB images are not used by the policy.

**Robotics Focus:**

4

**Summary Of Paper:**

This paper tackles the problem of learning a pick-and-place task for a quadrupedal robot with an arm. The task is accomplished with a high-level RL policy trained with a teacher-student method that outputs end-effector commands and body velocities, and a low-level policy that tracks these commands. Training is performed just in simulation.

**Summary Of Recommendation:**

I gave a high ranking because I think that the authors propose an interesting and novel method with convincing experimental validation.

---

### Official Review · Reviewer_KPpy · 2024-07-20
**Strong Accept**

**Originality:** 4
**Technical Quality:** 5
**Clarity Of Presentation:** 5
**Potential Impact:** 4
**Recommendation:** 4
**Confidence:** 5

**Review:**

The work demonstrated transfer of simulation trained picking policies, which can operate fully autonomously in real. While not entirely novel (prior works can also do autonomous pick and place), their system does not assume a Boston Dynamics-level team of engineers to calibrate the model and tune the controller, greatly improving accessibility of quadruped manipulation research to new hardware platforms. Further, their formulation of using RL policies enables (1) more elegant behavior thanks to the lack of separation between a walking and manipulation phase (seen in the supplementary video) and (2) the discovery of robust recovery behavior.

Overall, VBC is an impressive system, with many efforts to handle timing accuracies, latencies, and model mismatches. It includes a thorough experiments and description of evaluation setups in both simulation and real.



Small points:
- "Average Success Times v.s. Trial Times" is a bit confusing in Figure 7. Consider changing it to just "Success Rate v.s. Attempt"?
- There are many typos across the paper. Some examples include "previlleged" (Figure 3), "privleged"  (Table 1).

**Quality Of The Limitations Section:**

3

**Questions For Rebuttal:**

- 1. The work proposed to train a RL-based high-level policy, and have put efforts into tuning the RL set up for pick up task. Could the authors expand on the scalability of using RL for learning new tasks, versus other high-level policy approaches, such as trajectory planning or behavior cloning?
- 2. Using object masks is a very robust way to transfer to real, since as segmentation and tracking tools become very robust. However, this design choice, while sufficient for object picking, may corrupt information that's crucial for other task. Could the authors expand their limitation section on the applicability of this design choice to other tasks?
- 3. I find it interesting that IK worked well for the author's applications, given that DeepWBC tried and reported much worse numbers for the MPC+IK baseline, where one common failure mode in their case were body arm collisions. Is this sidestepped in the author's approach by a better hardware design (mounting the arm closer to the front), or replacing MPC with RL trained WBC (to avoid collisions in training)?

**Robotics Focus:**

4

**Summary Of Paper:**

This paper proposes a bi-level approach for achieving fully-autonomous real world vision-based grasping on a quadruped manipulator platform, trained using RL in real.

**Summary Of Recommendation:**

The paper demonstrates novel fully-autonomous capabilities, contains interesting details across the stack for achieving good manipulation object generalization and Sim2Real, has a thorough evaluation in both simulation and real, and is well written.

---

### Official Review · Reviewer_FL2x · 2024-07-26
**This paper proposes Visual Whole-Body Control (VBC) that does whole body control with visual observations. VBC utilizes three phases of training: 1) training of a universal low-level policy using RL to track goals and demonstrate whole whole-body behaviors, 2) training a privileged teacher policy using RL to complete a task, and 3) distilling teacher policy in to depth-image based student policy via online imitation learning. All training has been completed simulation and zero-shot deployment on real word with zero human-collected data.  Extensive simulations demonstrate its effectiveness in picking up diverse objects across various heights. Real-world experiments showcase successful pick-up tasks on unseen objects with emergent retrying behaviors.**

**Originality:** 3
**Technical Quality:** 5
**Clarity Of Presentation:** 5
**Potential Impact:** 4
**Recommendation:** 4
**Confidence:** 4

**Review:**

The paper is well-structured and easy to follow. The authors have included a thorough appendix with additional details and a comprehensive review of related work.

Pros:

+  VBC tackles the complex loco-manipulation problem by dividing it into manageable sub-problems, allowing for effective training and generalization.
+ The focus on using visual observations (segmented depth images) makes the system more autonomous and adaptable to real-world environments.
+ Successful deployment of the trained policy on a real robot with zero real-world data collection signifies the effectiveness of the proposed training pipeline.
+ The system demonstrates an emergent capability to retry grasping attempts when initial attempts fail, increasing robustness and success rates.
+ Extensive simulation experiments across various object categories and heights, alongside real-world tests on unseen objects, provide strong evidence for the framework's effectiveness.

Cons:

- While the pick-up task is convincingly addressed, the framework's applicability to other, more complex manipulation tasks remains unexplored.
- The reliance on a beak-like gripper limits the range of objects that can be manipulated and necessitates pushing objects instead of grasping.
- The framework's performance is highly dependent on accurate depth estimation, making it sensitive to errors in depth perception.
- The visual tracking system used can be prone to losing track of the object, especially in cluttered environments or with similar-colored objects.
- The hierarchical architecture with asynchronous multi-processing and different control frequencies might lead to computational overhead, potentially impacting real-time performance.

Areas for potential improvements:

* Evaluating VBC on a broader range of manipulation tasks beyond pick-up, such as pushing, placing, and opening, would demonstrate its wider applicability.
* Replacing the beak-like gripper with a parallel one would allow for more versatile grasping and improve manipulation accuracy.
* Investigating and integrating more robust and precise depth estimation methods, potentially combining RGB and depth information, could enhance performance.
* Exploring more robust visual tracking algorithms or incorporating multimodal perception, such as tactile sensing, could address tracking limitations.
* Further optimizing the system's computational efficiency, potentially by streamlining the hierarchical structure, would improve real-time control capabilities.

**Quality Of The Limitations Section:**

2

**Questions For Rebuttal:**

* Line 110 => rad/s
* Line 110 => “task-planning policy and further clipped into smaller ranges”. What is the reason for the discrepancy in size? To what extent is the size difference? The appendix does not contain the information in question.
* Line 114 => yaw
* Line 117 => While the environment extrinsic vector has been introduced, its application in conjunction with low-level hardware controllers remains unclear. Appendix C.1 fails to provide adequate explanation regarding this aspect. Given the crucial role of the extrinsic vector in sim-to-real transfer, it would be greatly appreciated if you could provide a detailed explanation or direct me to a specific reference that addresses this topic.
* Line 128 => wondering whether the authors have experimented with the damped least squares. Provide a reference, e.g., https://www.cs.cmu.edu/~15464-s13/lectures/lecture6/iksurvey.pdf.
* Line 165-169: “Moreover,  …”. The sim-to-real transfer strategy mentioned in this document requires further clarification. Kindly provide a more detailed explanation and ensure that any verbose information is moved to an appendix.
* Line 182: Type after a_{t-1}.
* Line 185: In the appendix, there is a reference to DAgger. Kindly provide that reference.


* One of the fundamental presumptions in training and deployment is that the objects of interest must be visible within the viewport. If this assumption is not met, can the methodology still function effectively?
* While the paper employs the term "autonomous," the methodology is technically "semi-autonomous." A human user must supply an augmentation mask and ensure the robot faces the object. To accurately reflect this, consider removing the word "autonomous" or adding the adjective "semi-" before it.
* Fig. 6. Is the labeling of minimum and maximum values correct? The description mentions two baselines, but the results only show one baseline. This inconsistency should be addressed.
* What are the primary factors contributing to the average simulation success rate hovering around 60%? What specific strategies or techniques can be implemented to enhance the simulation success rate beyond the current 60% average?
* Table 1: please provide the standard error.
* The experimental setup in the real world seems to be confined to a singular environment. Although Fig. 1 depicts activities in various environments, the results do not validate this. Thus, Fig. 1 can be considered misleading, particularly regarding the stringent requirements imposed on TrackingSAM and the intricate nature of task completion.
* Is the implementation based solely on tabular rasa, or have other libraries been utilized? If other libraries have been incorporated, kindly provide the required attributions for acknowledgment.
* Appendix B.4: please provide the references.

**Robotics Focus:**

4

**Summary Of Paper:**

The authors propose VBC, a three-phase cycle that maximally utilizes sim-to-real to zero-shot transfer policies to hardware. The main idea revolves around learning a generalized low-level controller that takes high-level goals, and a high-level controller that provides goals to complete a task. The novelty of the approach is twofold: 1) distilling the high-level controller to a visual observation, emerging recovery behaviors, and improving results compared to a static height, and 2) training all tasks in sim and zero-shot transferring them to reality. The methodology does not constrain itself with real-world constraints to define tasks and collect data, therefore, theoretically scaling for many tasks. The downside of the methodology is sim training and hardware deployment, as the objects of interest need to be in the view of the robot. There are no ablations if that assumption is relaxed and the methodology would work. The authors have addressed these limitations. Though Figure 1 shows the methodology worked on different environments, it is not experimentally validated.

**Summary Of Recommendation:**

The paper presents a vision-based control framework for legged loco-manipulation. The hierarchical approach, successful sim-to-real transfer, and emergent retrying behavior are key strengths. VBC has potential for future research and development in whole-body control, but there are limitations in gripper design, depth estimation, and tracking robustness.

---

### Author Rebuttal · Authors · 2024-08-14

## General Response and Manuscript Changes:
We thank all the reviewers for their valuable and constructive comments.

Based on reviewers’ feedback and suggestions, we made efforts to revise the draft (changes are highlighted in red) and related materials to make it more self-contained. **Please see our updated manuscript and videos in the attachment for details.** And we summarize the major of them below.

### **Additional Experiments**
- [In response to Reviewer xf7y] **Runtime analysis** to show how exactly all components coherently work together on the real robot, in Appendix F.


### **Additional Details**
- [In response to Reviewer FL2x] Key information of the low-level task space, like the frames of commands, how they are sampled, the exact range, and so on, in Section 3.1 and Appendix C.1.
- [In response to Reviewer FL2x, xf7y]  More details of regularized online adaptation (ROA) with an illustrative figure in Appendix C.1.
- [In response to Reviewer KPpy] Summary of domain randomization for low-level and high-level parts, in Appendix D.
- [In response to Reviewer Xf7y] More details about the model used for object annotation and segmentation mask generation, are in Appendix I.
- [In response to Reviewer FL2x, KPpy] Extended limitations discussing the beak-like gripper, the visual inputs, and the depth estimation, in Appendix H.
- [In response to Reviewer xf7y] We uploaded a video to illustrate how TrackingSAM works in our real robot experiment.
- [In response to Reviewer xf7y] The main demo video is updated with additional captions noting that the RGB images are not inputs

Besides, we tried our best to **fix inaccurate statements and improve readability** by splitting long sentences. We commit to polishing our writing and fixing as many typos as possible.

In the responses to each reviewer, we carefully address every comment and provide detailed explanations. We are willing to discuss specific issues further.

---

### Decision · Program_Chairs · 2024-09-04

**Decision:**

Accept

**Comment:**

The submission presents a new sim2real method for controlling locomotion and manipulation in parallel, leading to strong results and a detailed evaluation.

The submission effectively breaks down loco-manipulation problems into sub-problems, uses visual observations (segmented depth images), demonstrates successful deployment of the trained policy on a physical robot without requiring real-world data, robust retrying grasping attempts when initial attempts fail, enhancing overall success rates, is rigorously tested through broad simulation experiments and real-world tests on unseen objects.

While effectively addressing a pick-up task, applicability to more intricate manipulation tasks remains unexplored. The reliance on a beak-like gripper restricts the range of objects that can be manipulated, necessitating pushing instead of grasping. Performance heavily relies on accurate depth estimation. More details are needed about the model used for object annotation and segmentation mask generation, computation time of each module. The paper's readability is limited by long paragraphs.

The paper has improved during the rebuttal via detailed communication between reviewers and authors. Based on the above the paper is recommended for acceptance. Please follow up on the remaining open points for a potential camera ready version.